

# Multi-proxy speleothem-based reconstruction of mid-MIS 3 climate in South Africa

Jenny Maccali[1,2], Anna Nele Meckler[1,2], Stein-Erik Lauritzen[1,2], Torill Brekken[3], Helen Aase Rokkan[3], Alvaro Fernandez[4], Yves Krüger[3], Jane Adigun[5], Stéphane Affolter[6], Markus Leuenberger[7]

[1]Department of Earth Sciences and Bjerknes Centre for Climate Research, University of Bergen, Bergen, N-5007, Norway
[2]SFF Centre for Early Sapiens Behaviour (SapienCE), University of Bergen, Bergen, N-5020, Norway
[3]Department of Earth Sciences, University of Bergen, Bergen, N-5007, Norway
[4]Andalusian Institute of Earth Scienecs, University of Granada, Granada, 18100, Spain
[5]Department of Anthropology and Archaeology, University of South Africa, Pretoria, 0002, South Africa
[6]Department of Environmental Sciences, University of Basel, Basel, 4056, Switzerland
[7]Climate and Environmental Physics Division, Physics Institute and Oeschger Centre for Climate Change Research, University of Bern, Bern, 3012, Switzerland

*Correspondence to*: Jenny Maccali (jenny.maccali@uib.no)

**Abstract.** The southern coast of South Africa displays a highly dynamical climate as it is at the convergence of both the Atlantic and Indian Ocean, and it is located near the subtropical/temperate zone boundary with seasonal influence of easterlies/westerlies. The region hosts some key archeological sites with records of significant cognitive, technological and social developments. Reconstructions of the state and variability of past climate and environmental conditions around sites of archeological significance can provide crucial context for understanding the evolution of early humans. Here we present a short but high-resolution record of hydroclimate and temperature in South Africa. Our reconstructions are based on trace elements, calcite and fluid inclusion stable isotopes, and fluid inclusion microthermometry from a speleothem collected in Bloukrantz Cave, in the De Hoop Nature Reserve in the Western Cape region of South Africa.

Our record covers the time period from 48.3 to 45.2 ka during Marine Isotope Stage 3. Both $\delta 18Oc$ and $\delta 13Cc$ show strong variability and covary with Sr/Ca. This correlation suggests that the control on these proxies originates from internal cave processes such as Prior Calcite Precipitation, which we infer to be related to precipitation amount. The hydroclimate indicators furthermore suggest a shift towards overall drier conditions after 46 ka, coincident with a cooling in Antarctica and drier conditions in the eastern part of South Africa corresponding to the Summer Rainfall Zone.

Fluid inclusion-based temperature reconstructions show good agreement between the oxygen isotope and microthermometry methods, and results from the latter display little variation throughout the record, with reconstructed temperatures close to the present-day cave temperature of 17.5 °C. Overall, the BL3 record thus suggests stable temperature from 48.3 to 45.2 ka whereas precipitation was variable with marked drier episodes on sub-millennial timescales.


## 1 Introduction

The southern region of South Africa is a key region for the study of human evolution. Homo sapiens was anatomically modern as early as 300,000 years ago, however, no evidence for 'modern' behavior at that time can be inferred from
behavioral proxies in the archeological record (Hublin et al., 2017; Högberg and Lombard, 2021). Episodes of significant cultural changes, seemingly not related to subsistence requirements, have been identified during the Middle Stone Age (MSA) with artifacts such as engraved ochre, an ochre processing kit, engraved ostrich eggshells, bifacial points, and perforated marine shell beads (Henshilwood et al., 2011; e.g. Henshilwood et al., 2014; Marean et al., 2007; Wurz, 2002). The environmental conditions at the time have been suggested to drive changes either by offering refuge (suitable habitat)
allowing for experimentation, or on the contrary, by forcing innovative mechanisms of adaptation (D'errico, 2003; Wadley, 2021). However, although the number of paleoenvironmental reconstructions in South Africa's southern Cape coastal region during the MSA is increasing (e.g. Bar-Matthews et al., 2010; Braun et al., 2019b; Braun et al., 2020; Chase, 2010; Strobel et al., 2022), information on past climate in this region still remains sparse. In this study, we apply a suite of traditional and novel approaches to reconstruct climate in South Africa from a speleothem from Bloukrantz cave, that grew during a short
interval during Marine Isotope Stage (MIS) 3. Our goal is to cross-validate the various proxies and reconstruct the state and variability of hydroclimate and temperature.

Today, South Africa's climate is marked by different seasonal and spatial rainfall patterns (Fig. 1). During austral winter, the southern westerlies wind belt is displaced northward, bringing precipitation to the southwestern tip of South Africa (the Winter Rainfall Zone – WRZ). During austral summer, the westerlies are displaced southwards, allowing easterlies to bring
rain to the eastern part of South Africa (the Summer Rainfall Zone – SRZ). The junction of these two rain zones is known as the Year-round Rainfall Zone (YRZ), with winter rain accounting for 30 – 60 % of the annual precipitation and no marked seasonality (Carr et al., 2006; Chase and Meadows, 2007; Roffe et al., 2019). Identifying the drivers of climate change in the YRZ is not straightforward as the YRZ is a transition zone between WRZ and SRZ and thus influenced by a variety of mechanisms: Indian Ocean SST, convective and tropical weather systems in the east (Engelbrecht et al., 2015), and the
westerlies belt position and intensity, along with the associated frontal systems, in the west (Chase, 2010; Chase and Meadows, 2007).

MIS 3, the period between ~ 60 to 29 ka during the last glacial period, is characterized by a global mean sea level lower than today (Siddall et al., 2008) and globally colder temperature (Van Meerbeeck et al., 2009). Northern hemisphere ice core proxies reveal substantial millennial-scale variability such as Dansgaard-Oeschger and Heinrich events (Andersen et al.,
2004) and associated temperature changes (Huber et al., 2006; Kindler et al., 2014). In the Southern Hemisphere, Antarctic ice core records display similar variability, though of a lesser amplitude (Siddall et al., 2008). The Antarctic ice core record is closely mimicked by sea surface temperature (SST) reconstructions from sediment cores surrounding the southern tip of the African continent, both in the Indian (e.g. Simon et al., 2013) and the Atlantic sector (e.g. Dyez et al., 2014; Peeters et al., 2004). Mean annual precipitation reconstructions in the SRZ of Southern Africa closely follow solar radiation intensity
with reduced amplitude variation during MIS 3 compared to MIS 4 and 5 (Partridge et al., 1997). This is illustrated in a spelelothem sample from Wolkberg cave, where a drying trend was recorded from ~51 to 46 ka and linked to decreasing



solar radiation (Holzkämper et al., 2009). In the YRZ, fynbos pollen numbers indicate a drying period from 60 to 40 ka followed by wetter conditions from 40 to 30 ka (Quick et al., 2016). In the speleothem record, this is illustrated by a marked decrease in the overall number of speleothem samples recovered at ~60 ka, followed by a slight increase from 45 to 30 ka

(Braun et al., 2019a). Moreover, a speleothem from the Little Karoo region displays trends similar to terrestrial runoff from the Namibian west coast, suggesting a dominant contribution of winter rains (Braun et al., 2020). Finally, aridity index reconstructions indicates variable aridity conditions in the WRZ through MIS 3, although with overall drier conditions compared to MIS 4 (Stuut et al., 2002).

Speleothems are cave deposits (most often Ca carbonates), which can be accurately dated by the U-Th method. They are

most commonly used to reconstruct changes in precipitation on the basis of variations in the oxygen isotopic composition ($\delta$18O) of the calcite matrix (Lachniet, 2009). The carbon isotopic composition ($\delta$13C) of speleothem calcite is more complex to interpret, as it can reflect changes in vegetation (C3 vs C4 plants) above the cave and/or cave internal processes leading to C isotope fractionation (Fohlmeister et al., 2020). The latter are commonly also reflected in variations in trace element to Ca ratios, such as Mg/Ca or Sr/Ca (Stoll et al., 2012). Here we combine both $\delta$18O and $\delta$13C from the calcite

(later noted with subscript c), and Sr/Ca ratios to infer about past change in precipitation.

Recently, quantitative proxies for cave temperature have been developed (Affek et al., 2008; Blyth and Schouten, 2013; Kluge et al., 2008; Krüger et al., 2011; Vonhof et al., 2006). Cave temperature generally reflects the mean annual air temperature outside of the cave (Poulson and White, 1969), making cave deposits ideal candidates for land temperature reconstructions. The first quantitative temperature reconstruction method that has been proposed is the water-carbonate

paleothermometer based on oxygen isotopes. The theoretical background of this approach dates back to the 1960s (Epstein et al., 1951; Epstein et al., 1953; Mccrea, 1950; Urey, 1947). In speleothems, however, the application of this thermometer has initially been limited by the lack of knowledge of the water isotopic composition. This information can now be gained from fluid inclusion water isotope (FIWI) measurements (e.g. Affolter et al., 2014; Fernandez et al., 2023; Matthews et al., 2021; Vonhof et al., 2006; Warken et al., 2022; Wassenburg et al., 2021; Wortham et al., 2022), which reveal the isotopic

composition of former drip water preserved in microscopic inclusions in the speleothem calcite. Here we combine temperature estimates based on the difference in oxygen isotopic composition of fluid inclusions and calcite with another, independent temperature proxy, namely fluid inclusion microthermometry, which uses liquid-vapor homogenization temperatures to determine the density of the enclosed drip water (Krüger et al., 2011; Løland et al., 2022). The suite of all methods applied here allows us to derive multi-proxy records of both hydroclimate and temperature.

**2 Material and Methods**

**2.1 Site description and setting**

Bloukrantz cave (34°27.557'S, 20°46.697'E, 10-25 masl) is located along the coast of South Africa in the De Hoop Nature reserve in the southern Cape region (Noah, 2011). The cave is a composite marine abrasion cave formed in quartzite where the entrance is almost completely closed by travertine derived from overlying, aeolian calcarenite dunes. The narrow





entrance leads to a first chamber followed by a steep slope down to the main room (Fig. S1). The interior of the cave is largely filled with columnar stalagmites that have grown since the entrance wall closed the cave. The speleothem used in this study (BL3) was collected in a smaller chamber adjacent to the main room (Adigun, 2016). The cave floor mainly consists of sand mixed with bat guano. An Onset HOBO U23-001 ProV2 temperature logger was placed in the cave in February 2018 and data were collected in January 2019 and March 2020. In 2018, dripping was not active, and the logged relative humidity

(rH) was ~90%. During the two subsequent visits in January 2019 and March 2020, dripping in the cave was active and the logged rH was ~100% (Fig. S2). Temperature in the cave was fairly stable between February 2018 and March 2020 and varied from 16.4 to 18.8 °C with a mean annual temperature of 17.5 ± 0.5°C. Slightly further inland, at the Potberg station (34°22.623'S, 20°02.044'E, 176 masl), the mean annual temperature was 16 ± 5 °C for the same period with annual precipitation of 220-380 mm. At Klipdrift sea cave (34°27.096'S, 20°43.458'E), a few kilometers west along the coast, a

mean annual temperature of 17.6 ± 0.3 °C was recorded. The similar temperatures recorded at the two sites allow us to exclude potential warming from guano degradation at Bloukrantz cave as the Klipdrift sea cave does not shelter a bat colony. Bloukrantz cave is ideally positioned in the YRZ to provide local paleoclimate reconstructions in relation with key archeological sites such as Blombos cave and Klipdrift shelter.

## 2.2 Sample description

Sample BL3 (Figs. 2, S3) is 425 mm long and 105 mm wide at its widest (close to mid-height). The stalagmite displays two distinct growth episodes with a clear hiatus at 198 mm (Fig. S3). The pre-hiatus part consists of white milky calcite with microcrystalline fabric (Frisia, 2015) and displays multiple changes of the direction of the stalagmite growth axis, most likely linked to lateral shifts of the dripping site. After the hiatus, stalagmite BL3 features a 100 mm long straight section of translucent calcite with brittle columnar fabric (Frisia, 2015) that also covers the flanks of the lower part. The top 95 mm

consist again of microcrystalline, milky calcite, and shows clear layering.
In this study, we are focusing mainly on the lower part of BL3 that formed during MIS 3. Dark layers and clear growth axis changes can be observed at 241, 292, 312, 354, 380, and 401 mm (dashed lines in Fig. 2). The surfaces of these dark layers show slight dissolution features and indicate potential short-term growth interruptions of the stalagmite (S. Frisia, personal communication).

Apart from these dark hiatus layers, inspections of thin sections did not reveal any significant changes of the calcite fabric throughout the MIS 3 part of the sample. The orientation of the calcite crystals in the microcrystalline fabric does not exhibit preferential crystallographic orientation, which is indicated by the tipped terminations of small intra-crystalline fluid inclusions. The calcite fabric as a whole is quite porous, which explains its milky appearance. Fluid inclusions, both intra- and inter-crystalline are abundant but of small size. A large portion of the inclusions is two-phase containing liquid water

and a gas bubble. It is not yet clear whether the gas bubble contains air that might have been trapped during the formation of inclusions, or water vapor that would rather indicate post-formation water loss or volume alteration of the inclusions. Mono-phase liquid inclusions, in contrast, were found to be relatively sparse.



### 2.3 Sample preparation

The sample was cut lengthwise into 1 cm thick slabs. One slab was used for X-ray Fluorescence (XRF) scanning while a
second slab cut form the other half of the stalagmite was used for dating, isotopic analyses, and microthermometry analyses. Since stable isotopes and trace elements were measured on different slabs, the Sr/Ca and the stable isotopes transects show slight offsets which can be accounted for by tracing visible layers in both slabs (grey bars in Fig. 2 connect equivalent features).

### 2.3.1 U-Th dating

Subsamples for dating were drilled using a Sherline 5410 milling stage mounted with a 1.5 mm drill bit. The chemical separation procedure was largely derived from Edwards (1988). Briefly, ~250 µg of carbonate powder was spiked using a mixed solution of $^{229}$Th-$^{233}$U-$^{236}$U, calibrated using a Harwell uraninite (HU-1) solution considered at secular equilibrium. After dissolution with concentrated $HNO_3$, Fe-precipitates were formed by addition of clean Fe and stepwise addition of $NH_4OH$. After centrifugation the Fe-precipitate was dissolved in HCl and loaded onto AG1X8 resin, where Th was separated
from U. Each fraction was then purified by another pass through AG1X8 resin for Th and U-TEVA resin for U. Isotopic ratios were measured at the Department of Earth sciences at the University of Bergen in dry plasma mode on a Nu Plasma II instrument upgraded with a plasma 3 source. Isotopic ratios were measured by peak jumping on a secondary electron multiplier (SEM). Mass bias was corrected using the $^{236}$U/$^{233}$U spike ratio. A HU-1 solution was used as a standard solution to monitor analytical sessions. Activity ratios were calculated using decay constant values from Cheng et al. (2013). Ages
were calculated using the Excel Isoplot add-in 3.75 (Ludwig, 2003) without decay constant uncertainties. All U-series data reported in tables and figures are presented with a ±2σ uncertainty. The $^{232}$Th/$^{238}$U bulk Earth ratio of 3.8 was used to correct $^{230}$Th ages for detrital Th contamination (Taylor and Mclennan, 1985). Two samples were additionally dated at the Isotope Laboratory at Xi'an Jiaotong University (see supplement).

### 2.3.2 Trace elements

Sr/Ca ratios can be rapidly obtained by non-destructive XRF scanning (Scroxton et al., 2018). For this study, Sr/Ca ratios were measured on an ITRAX XRF core scanner from Cox Analytical Systems (Gothenburg, Sweden) using a 3 kW molybdenum (Mo) X-ray tube. The voltage was set to 28 kV, current to 28 mA, resolution to 200 µm, and exposure time to 20 s (Rokkan, 2019).

### 2.3.3 Stable isotopes

Using the milling stage, carbonate powder was milled continuously in 1mm increments along transects following the growth axis of the BL3. Oxygen and carbon isotope ratios were measured on 30-50 µg samples following routine protocols at Farlab (Facility for advanced isotopic research and monitoring of weather, climate and biogeochemical cycling) on a Thermo Fisher Scientific MAT253 isotope ratio mass spectrometer with a Kiel IV carbonate preparation device. The δ$^{13}$C$_c$- and δ$^{18}$O$_c$-





values (subscript c stands for calcite) were calibrated against an in-house marble standard and NBS18, and are expressed in
‰ against VPDB. Reproducibility of standard measurements was better than 0.10 ‰ (1σ) for $\delta^{18}O$ and better than 0.05 ‰
(1σ) for $\delta^{13}C$.

### 2.3.4 Microthermometry (liquid-vapor homogenization temperature)

The microthermometric approach uses the density of water in stalagmite fluid inclusions as a proxy to reconstruct cave
temperature. The application of the microthermometry method to fluid inclusions in stalagmites is described in detail by
Krüger et al. (2011) and sample preparation is described in Løland et al. (2022). Briefly, blocks of 20 mm width and 30-40
mm length were cut from the second slab alongside the isotope transects. Then, ~300 µm thick sections were cut from the
calcite block with a low-speed saw (Buehler Isomet), and these unpolished thick sections were broken into smaller pieces of
~4x4 mm to fit on the sample holder of the microscope heating/freezing stage (Linkam THMS600). Individual fluid
monophase inclusions were selected for analysis and cooled to 5 °C. At this temperature the inclusion water is in a
metastable liquid state and a femtosecond laser pulse was used to nucleate a vapor bubble. The liquid-vapor homogenization
temperatures ($T_{h(obs)}$) were subsequently recorded when the vapor bubble disappeared upon controlled heating of the sample.
$T_{h(obs)}$ was corrected for surface tension effects using the vapor bubble radius measured at known temperature and a
thermodynamic model of Marti et al. (2012) to calculate the water density and thus the formation temperature of the fluid
inclusion. Information on the original density of the former drip water can be obtained only from mono-phase liquid fluid
inclusions. Measurements of $T_{h(obs)}$ and of the bubble radii were challenging because of the small size of the inclusions (100 -
3000 µm$^3$). In some cases, the collapse of the vapor bubble at $T_{h(obs)}$ could not be observed directly. In these situations, a
temperature cycling procedure with stepwise heating and subsequent cooling was applied to determine the homogenization
temperature precisely. Bubble images for the radius measurements were taken at 5.1 ˚C where the vapor bubble in a calcite
confined system reaches maximum size. Mean temperatures of coeval inclusions from the same growth layers were
185    considered as a best estimate of the stalagmite formation temperature at the respective sample position. Temperature
uncertainties are reported as 2σ standard error of the mean.

### 2.3.5 Fluid inclusion water isotopes

The remaining part of the blocks was divided in 3-5 mm wide lamella cut along the curved stalagmite growth layers using a
diamond wire saw (Well 3421). These layer-parallel samples were then split into coeval subsamples of about 0.2-0.4 g for
190    replicate measurements of fluid inclusion water isotopes. A total of 31 layers were analyzed, 25 at Farlab in Bergen and 6 at
the University of Bern. The analytical setup in Bergen is described by Sodemann et al. (In Prep). Briefly, aliquots of >100
mg were crushed in a heated (120 °C) crusher device (similar to that described by e.g. De Graaf et al., 2020) connected to a
Picarro L2130-i laser spectrometer. A microdrop device ensures a stable humidity background in the air stream that purges
the crusher. After loading the sample into the preheated crusher, it took about 15-20 minutes to achieve a stable water
195    background in the system. The fluid inclusion water was then released by crushing the sample and its isotopic composition



was determined by subtracting the water background from the signal (Affolter et al., 2014). The analytical setup in Bern is described in Affolter et al. (2014).

FIWI analyses could be performed only in the MIS 3 part of the stalagmite and on the topmost layers, because water yields from the columnar fabric of the Holocene part were too low. Data accuracy and reproducibility were estimated using in-house water standards sealed in borosilicate capillaries and crushed in the analytical line, and water standard injections. Reproducibility was <0.4 ‰ for $\delta^{18}O$ and <1.2 ‰ for $\delta^2H$ (± 1σ). Results are reported as the average of 3 replicates and uncertainties are calculated as 1σ standard deviation or set as 0.4 ‰ for $\delta^{18}O_w$ and 1.2 ‰ for $\delta^2H_w$, whichever was larger. FIWI temperatures were calculated using the empirical relationship from Tremaine et al. (2011), with the $\delta^{18}O_c$ measured on the crushed carbonate remaining after water isotope analyses. Uncertainties are reported as ±1 σ and include error propagation of both water and calcite $\delta^{18}O$.

## 3 Results

### 3.1 U-Th dating and Age model

A total of 21 dates were obtained and range from 1.29 ± 0.01 to 47.54 ± 0.37 ka (see supplementary material). Two dates (at 428 and 339 mm) were rejected as clear outliers. The age-depth model was calculated using the StalAge algorithm in R (Scholz and Hoffmann, 2011) as two distinct sections, before and after the hiatus, and is reported with a 95% confidence interval. The two dates performed at the Isotope Laboratory at Xi'an Jiaotong University have been included in the age-depth model. The age-depth model (Fig. S4) displays an almost linear growth from 48.4 to 45.2 ka with an average growth rate of 0.07 mm/a except from ~46.0 to 46.4 ka when the growth appeared to have been faster with an average growth rate of 0.19 mm/a. After the hiatus, the growth was slower from 7.6 to 3.6 ka (0.06 mm/a) and faster from 3.6 to 1.5 ka (0.09 mm/a).

### 3.2 Trace elements

The Sr/Ca count ratios range from 288 to 687 (Fig. 2). For the MIS 3 section, a series of positive excursions of variable amplitude are overprinted on the baseline signal. The baseline increases slightly from 48.4 to 46 ka, followed by a more pronounced increase after 46 ka.

After the hiatus, the Sr/Ca signal drops markedly and shows little variation with an average value of 348, followed by a gradual increase towards the top of the stalagmite. The Holocene section presents an overall higher frequency variability.

### 3.3 Stable isotopes

MIS 3 and Holocene $\delta^{13}C_c$ values range from -7.9 to -1.4 ‰ and from -8.6 to -2.6 ‰, respectively. $\delta^{18}O_c$ values range from -3.7 to -1.0 ‰ for the MIS 3 part and from -5.1 to -2.6 ‰ in the Holocene section. $\delta^{18}O_c$ and $\delta^{13}C_c$ closely follow the same pattern. As for the Sr/Ca record, the isotopic baseline of the MIS 3 section displays little variation from 48.4 to 46 ka followed by an increase after 46 ka of up to 3 ‰ in $\delta^{13}C_c$ and 1 ‰ in $\delta^{18}O_c$. A series of peaks is superimposed on the baseline in both the $\delta^{13}C_c$ and $\delta^{18}O_c$ records and corresponds to similar peaks in the Sr/Ca signal and the presence of dark





layers in the sample. The amplitude of these excursions varies from 2.3 to 5.1 ‰ for $\delta^{13}C_c$ and from 0.9 to 1.7 ‰ for $\delta^{18}O_c$. The Holocene part displays lower values in both $\delta^{13}C_c$ and $\delta^{18}O_c$ immediately after the hiatus with little variation until 3.5 ka, followed by a gradual increase until the top of the record at ~1.5 ka.

### 3.4 Microthermometry

A total of 17 layers were analyzed including one in the Holocene part. Between 5 to 20 replicates were performed for each layer depending on the number and size of inclusions. Most samples display skewed (towards either low or high values) normal-like distributions with an absolute range among individual measurements from the same layer ranging between 3-6 °C (see Fig. S5). Some samples showed a larger range (7-9 °C) with semi-uniform (i.e., flatter) distributions and larger standard error of the mean; nonetheless, these samples provided mean temperatures similar to adjacent samples with smaller ranges (Fig. S5). This translates into standard errors of the mean ranging from 0.5 to 2.0 °C. The larger errors compared to previous studies (e.g., Løland et al., 2022) are due to the limited number of measurable inclusions in BL3. In general, the temperatures are very consistent throughout the stalagmite (Fig. 3). Stalagmite formation temperatures reconstructed from the topmost Holocene part of BL3 (microcrystalline fabric), dated at 1.8 ka, indicate an average value of 17.6 ± 0.6 °C (2σ), which is close to the present-day cave temperature (17.5 ± 0.5 °C) derived from cave monitoring. Temperatures from MIS 3 show little variation and range from 17.7 ± 1.1 to 20.7 ± 1.3 °C, with the highest temperature determined at ~45.8 ka (Fig. 3).

### 3.5 Fluid inclusion water isotopes

The FIWI data are distributed in two clusters (Fig. 4). The older cluster comprises samples from the base to ~46 ka and plots along the local meteoric water line (LMWL – from GNIP station at Cape Town airport from 1961 to 2013) with values ranging from -3.8 to -2.4 ‰ and from -17.4 to -10.2 ‰ for $\delta^{18}O_w$ and $\delta^2H_w$, respectively. Samples younger than ~46 ka plot as a distinct cluster, slightly off the LMWL, and with higher values ranging from -1.0 to -0.5 ‰ and from -4.7 to -2.0 ‰ for $\delta^{18}O_w$ and $\delta^2H_w$, respectively. Both $\delta^{18}O_w$ and $d^2H_w$ profiles display trends similar to the $\delta^{18}O_c$ baseline with little variations from 48.3 to 46 ka followed by an increase after 46 ka (Fig. 3).

There are three possible ways of calculating cave temperature based on the available dataset. The first method is by estimating today's $\delta^2H$ relationship to temperature and applying it to the past considering this relationship has not significantly changed over time (e.g. Affolter et al., 2019). At Mossel bay (east of Bloukrantz in the YRZ), Braun et al. (2017) found that there is a weak correlation (R=0.4) between $\delta^2H$ and temperature ($\delta^2H$=2.7(±0.6)*T-54(±10)) resulting in temperature estimates 4-5°C lower than microthermometry and standard deviation >4°C (1σ). The second approach is to calculate $\delta^{18}O_w$ from the measured $\delta^2H_w$ using modern $\delta^{18}O_w$ vs $\delta^2H_w$ relationship (i.e. LMWL), and then calculate the temperature using the Tremaine et al. (2011) equation (e.g. Meckler et al., 2015). This approach is often favored as $\delta^2H_w$ is less impacted than $\delta^{18}O_w$ by fractionation processes in the cave, it relies however on the assumption that the LMWL has not changed significantly over time. At Bloukrantz cave this results in FIWI temperature on average ~4°C lower than microthermometry and larger standard deviation (Fig. S6). The third approach is to calculate temperatures using the equation of Tremaine et al. (2011) with measured $\delta^{18}O_c$ and $\delta^{18}O_w$. This last approach is the one we selected as we believe it is the



most likely to render actual temperature variation in the cave. The calculated temperatures range from 15.4 to 21.1 °C from
48.3 to 46 ka and are in good agreement with liquid-vapor homogenization temperatures (Fig. 3 & S6) with the exception of
the peaks in $\delta^{18}O_c$ where temperatures are ~3°C colder. After 46 ka, FIWI temperatures depart from the microthermometry
results with positive offsets of 5 to 15 °C. d-excess values are fairly constant throughout the record with an average value of
9.7 ± 1.9, except for the younger samples that display decreasing values starting at ~46 ka (Fig. S7).

## 4 Discussion

### 4.1 Hydroclimate reconstructions

Interpreting isotopic and geochemical proxies in speleothems is not straightforward as epikarst/cave processes, directly or
indirectly linked to climate, can alter the proxy signals (e.g. Fairchild and Baker, 2012b; Mickler et al., 2004; Oster et al.,
2012). The BL3 record displays a strong correlation between $\delta^{18}O_c$ and $\delta^{13}C_c$ ($R^2$ values ≥ 0.9) that can reflect out-of-
equilibrium precipitation. Trace element incorporation (e.g., Sr) in the carbonate matrix is related to hydroclimate changes
and higher Sr/Ca values are commonly interpreted to reflect prior calcite precipitation - PCP (Baker et al., 1997; Fairchild et
al., 2000; Fairchild and Treble, 2009; Frisia et al., 2011; Wassenburg et al., 2020). PCP can be defined as calcite
precipitation upstream of the final drip site, either in the epikarst or in the cave itself, during i) periods of lower cave $pCO_2$
prompting $CO_2$ degassing and precipitation of calcite or ii) drier periods when an increased proportion of air in the lower
epikarst and/or longer residence time of the water on the cave ceiling/stalactites allow $CO_2$ degassing and precipitation along
the flow path. During PCP, many trace elements including Sr preferentially remain in the solution (Morse and Bender,
1990), appearing enriched over Ca in the subsequent calcite precipitating on the stalagmite. PCP also changes both $\delta^{13}C_c$ and
$\delta^{18}O_c$ towards higher values as light isotopes will be removed from the dissolved inorganic carbon (DIC) reservoir during
$CO_2$ degassing (Deininger et al., 2021; Dreybrodt, 2008; Hansen et al., 2019), with increases of up to 2 ‰ and 7 ‰,
respectively, for $\delta^{18}O_c$ and $\delta^{13}C_c$ at T=20°C (Hansen et al., 2019). A slope of 2.45 is estimated for the $\delta^{13}C_c$ vs $\delta^{18}O_c$
correlation from our dataset and points to incomplete O-isotope buffering between the DIC and $H_2O$ reservoir, based on the
Rayleigh distillation model developed by Mickler et al. (2006). In this model, a vertical slope corresponds to complete
buffering, while a slope of 0.52 is the theoretical limit for a system with no buffering. Drip-rate and cave $pCO_2$ are
considered to be the primary controls on PCP (Fohlmeister et al., 2020; Oster et al., 2012), with both lower cave $pCO_2$ and
lower drip-rate favoring PCP either in the lower epikarst or at the cave ceiling (Frisia et al., 2011). Individually, Sr/Ca ratios,
$\delta^{18}O_c$ and $\delta^{13}C_c$ can be influenced by a variety of mechanisms (e.g. Fairchild et al., 2000; Fohlmeister et al., 2020; Lachniet,
2009), however the correlation of the three proxies and the fact that the relative amplitude among peaks in both $\delta^{18}O_c$ and
$\delta^{13}C_c$ is similar indicate that these proxies are influenced by a common mechanism. We hence propose that stable isotope
and Sr/Ca ratios in stalagmite BL3 are controlled primarily by PCP.

Major growth direction changes and dark layers are concomitant with each peak and further examination of these layers
reveals signs of dissolution/erosion (pers. Comm. Silvia Frisia) that could correspond to short hiatuses and would indicate
that drip water availability rather than ventilation drives PCP in Bloukrantz cave. A likely scenario is therefore that periodic





drying episodes caused both the observed variations in the geochemical parameters and the visual changes in the speleothem. As conditions became drier, the drip-rate would have decreased, allowing for more PCP as for example during the period

with lower rH in 2018 when there was no active dripping in the cave (Fig. S2). Eventually, calcite growth would stop, allowing for dust to settle at the top (i.e. dark layers) and alteration of stalagmite surfaces, until growth resumed (with or without growth direction change) under wetter conditions.

Both the Sr/Ca ratio and the isotope profiles display an increase in the baseline after ~46 ka that indicates general drying if the same interpretation is applied (i.e. higher values reflect drier conditions). Interestingly, this increase in the baseline at

~46 ka corresponds to a thinning of the stalagmite's width (see Fig. 2) likely due to slower drip-rate as conditions became drier (Fairchild and Baker, 2012a). Overall, the record from 48.3-45.2 ka can thus be interpreted as variable precipitation from 48.3 to 45.2 ka with short, marked drier episodes and overall drying after ~46 ka. Based on our age model, the duration of the dry phases was ~ $200 \pm 200$ yrs, with relatively large uncertainty due to the $0.3 – 0.8$ % uncertainty of the U-Th dates. We also note that the duration could have been even shorter if unresolved hiatuses are present. Despite the remaining

uncertainty in the duration of the dry phases, it is clear that they represent processes operating on sub-millennial (centennial or decadal) timescales.

Comparison with other paleoclimate records is hampered by the scarcity of regional high-resolution records and by the relative short time period covered by our record (<3 kyrs). That being said, the Antarctic temperature record based on $\delta^{18}O$

from the EPICA Dronning Maud Land (EDML) ice core (Epica Community Members et al., 2006) shows some similarities. Notably, a cooling phase that starts at ~45.9 ka at EDML appears to coincide with what we interpret as overall drying at Bloukrantz cave (Fig. 5). Cooling in Antarctica has been associated with an equatorward shift of the southern westerlies belt, causing a northward extension of the winter rainfall zone along the west coast of Africa (Chase and Meadows, 2007; Stuut et al., 2002). This is illustrated in core MD96-2094 from Walvis Ridge off southwest Africa (19°59.97'S, 9°15.87'E) where the

Aridity index developed by Stuut et al. (2002) starts decreasing around 46 ka and, is interpreted as increased rainfall due to northward movement of the westerlies. Similarly, off Southeast Africa, on the Agulhas Plateau, an increase in ice-rafted debris at ~ 46.1 ka and a gradual decrease in Agulhas Leakage Fauna both on the Cape Basin record (Peeters et al., 2004) and in core CD 154 17-17K (Simon et al., 2013) are associated with a northward shift of the Subtropical Front. These observations have been interpreted as a northward shift of both atmospheric (southern westerlies belt) and oceanic

(subtropical front) circulation systems as a result of cooling in Antarctica. On land, a speleothem record from Wolkberg cave, in the Limpopo Province in the northeastern part of South Africa (Holzkämper et al., 2009), spanning the period of ~59 to 46 ka, displays a hiatus at 46.3 ka, coinciding with the onset of overall drier conditions at Bloukrantz cave. The presence of hiatus(es) is generally not systematically linked to drier conditions, however, other records offer some line of evidence for lower precipitations in the summer rainfall zone. The speleothem record from Lobatse cave in Botswana (Holmgren et al.,

1995) presents a sharp increase (~6 ‰) in $\delta^{13}C_c$ at ~46 ka followed by constant high $\delta^{13}C_c$ values and a hiatus at 43.2 ka. This signal was interpreted as drier conditions in the northeastern part of South Africa (in the SRZ). The growth period of these records collectively points to overall drier conditions in the Summer rainfall zone between 46 to 43 ka and correlates



well with reconstructed rainfall amount at Tswaing crater (Partridge et al., 1997) that shows a decrease in rainfall amount starting at 50 ka and reaching a minimum at 44 ka.

In combination with these lines of evidence from the SRZ, the overall drier conditions at Bloukrantz cave at 46 ka and the subsequent stop in growth at 45.3 ka could be explained by reduced summer rainfall through a northward shift of the southern westerlies belt. The drier conditions in the SRZ after 46 ka is therefore matched by wetter conditions in the WRZ as far north as 20°S, indicating the northward expansion of the WRZ at that time (Stuut et al. 2002). A direct implication would be that winter rainfall did not provide sufficient moisture to sustain carbonate growth at Bloukrantz cave and that the

northward expansion of the WRZ is not matched by a similar eastward expansion and/or is linked to reduced westerlies intensity. The inferred overall drying observed in our record and in records from the SRZ could reflect the potential influence of the Antarctic ice sheet through latitudinal displacement of the atmospheric circulation and the subsequent change in rain regime.

The repeated apparent drying we observed in our record illustrated by the isotope peaks is not matched by the isotopic record at Lobatse or Wolkberg cave. However, the Wolkberg record displays some marked variability in aragonite/calcite content with shifts from 100% calcite to >90% aragonite on sub-millennial to millennial timescales (Holzkämper et al., 2009). Holzkämper et al. (2009) tentatively linked higher aragonite content to drier conditions as factors controlling the formation of aragonite are low drip rates, higher temperature and high Mg concentration in the drip water, the latter likely linked to

decreased precipitation and longer residence time in the epikarst (Frisia et al., 2002). This could suggest that these sub-millennial events are not restricted to the southern Cape coastal area but may have been more regional.

### 4.1 Temperature reconstructions

The good agreement between the youngest microthermometry estimate with measured cave temperature (Fig. 3) shows that the microthermometry method can provide reliable cave temperatures for Bloukrantz cave, despite the challenges posed by

the small size of the fluid inclusions in BL3. Microthermometry temperatures are approximately constant throughout the MIS 3 part of the record with an average temperature of 18.8 ± 0.5 °C, i.e., slightly warmer compared to present day (Fig. 3c). The slightly warmer temperature is noteworthy given that the time period covered by the record is within the last glacial, with colder-than-Holocene temperatures in most parts of the world. Our results suggest that, in the southern Cape region of South Africa, the overall globally cooler conditions are offset by other influences, such as changes in ocean circulation or the

coastline distance due to lower relative sea level. During the time interval covered by our record, marine core CD 154 17-17K in the Indian Ocean (off the northeastern part of the Agulhas Plain - Simon et al. 2013) displays a decrease in SST, while marine core MD02-2594 in the Atlantic Ocean sector (off Cape Agulhas - Dyez et al., 2014) shows a slight increase in SST suggesting a potential larger influence of the South Atlantic Ocean on the southern Cape region. Moreover, a 70 m lower relative sea level at the beginning of our record (Grant et al., 2012) would have shifted the coastline by almost 10 km

(Jacobs et al., 2020). Göktürk et al. (2023) modeled that a coastline shift (~ -70 m at 70 ka) would result in drier conditions and more pronounced continentality along the coastline of the southern Cape region, with higher (lower) daily max (min)



temperature and overall higher mean annual temperature which could explain why higher than today temperatures are recorded at Bloukrantz cave.

Interestingly, no significant changes in temperature are found during the Sr/Ca and isotope peaks, suggesting that the process(es) influencing the calcite composition are not related to temperature. The peaks observed in the calcite-based proxies are also not apparent in the FIWI signal. When PCP occurs, $\delta^{18}O$ and $\delta^{13}C$ of the DIC increase as primary calcite is precipitated; the $\delta^{18}O$ of the DIC will then gradually decrease over time through equilibration with $H_2O$ (Deininger et al., 2021; Hansen et al., 2019). If the time between PCP and the subsequent calcite precipitation on the stalagmite is not long

enough to allow for O-isotope equilibration with $H_2O$, $\delta^{18}O_c$ of the stalagmite calcite will be elevated compared to what would be expected from the $\delta^{18}O_w$ and the cave temperature (Deininger et al., 2021; Dreybrodt and Fohlmeister, 2022; Hansen et al., 2019). Indeed, while FIWI temperatures calculated using the T-$\alpha$ relationship from Tremaine et al. (2011) show generally very good agreement with microthermometry from 48.3 to 46 ka, they deviate during the isotopic peaks with FIWI-T ~3°C colder than the corresponding microthermometry. The FIWI results thus further support our interpretation of

the isotope peaks as cave-internal processes controlled by hydroclimate.

After 46 ka, FIWI temperatures clearly depart from microthermometry with values 5 to 15°C warmer. In $\delta^{18}O_w$ vs $\delta^{2}H_w$ space, these younger samples plot as a distinct cluster away from the LMWL, in contrast to the samples older than ~46 ka (Fig. 4). Such departure from the LMWL has been observed earlier (Van Breukelen et al., 2008; Wainer et al., 2011; Warken et al., 2022) and could point either to analytical artefacts (e.g. Matthews et al., 2021) or to in-cave processes such as

evaporation (Warken et al., 2022). Water content in the samples can in some cases track potential water loss during the analytical procedure as fabric amenable to leaking will result in both lower water content and a departure from the MWL (Fernandez et al., 2023; Matthews et al., 2021). Here, the water content displays little variation through most of the record (Fig. S7) except for 2 samples with higher water content at ~45.9 ka, just before the FIWI data depart from the LMWL. In addition, replicate measurements of the younger samples do not show any trend in $\delta^{18}O_w$ vs $\delta^{2}H_w$ space as would be

expected from variable partial loss of water during heating of the samples (Fernandez et al., 2023). Further, no changes in the speleothem fabric were detected that could explain a change in behavior during analysis for these samples. We thus do not have any evidence that suggests analytical artefacts could cause the departure of the younger MIS 3 samples away from the LMWL.

An alternative explanation could be in-cave evaporation (e.g. Warken et al., 2022). Using a Craig-Gordon evaporation model

(Craig and Gordon, 1965) with a n value of 1 (i.e. non-turbulent atmosphere), and the average $\delta^{2}H_w$ and $\delta^{18}O_w$ values from the data points > 46 ka as a starting point, <5% loss to evaporation under rH between 80 to 85 % could explain the isotopic values of the younger samples. These are not unrealistic conditions, as rH of 86 % has been measured when no dripping was observed in the cave (Fig. S2). Cave evaporation occurs when relative humidity decreases as a consequence of i) better ventilation, when the cave air is partially replaced by outside air with a lower rH, or ii) lower drip rate, decreasing the water

supply to the cave and thus the rH. Wind-induced changes in ventilation seem unlikely given the cave geometry, whereas changes in ventilation induced by thermal convection (Fairchild and Baker, 2012c) are not supported by the apparently

constant microthermometry temperatures. We hence suggest that slower drip rate and lower water supply could be the cause for lower rH, leading to evaporation in the cave and elevated $\delta^{18}O_w$ and $\delta^2H_w$ of the younger samples at the end of the MIS 3 section. This interpretation is also in line with the calcite-based proxies suggesting a drying trend leading up to the prominent 400 growth hiatus.

In summary, multiple lines of evidence from our data suggest that temperature was stable throughout the record with an average value of 18.8 ± 0.5 °C, slightly higher than present-day temperature, whereas precipitation was variable with marked short drier episodes and overall drier conditions after 46 ka.

### 4.3 Significance for the archeological record

MIS 3 archeological sites display an undeniable change from the previous Howieson's Poort technological complex, with for instance fewer ornaments or decorated items (Cochrane, 2008; Mitchell, 2008; Wadley, 2015 and references therein). This change is also marked by a geographical shift from modern coastal (e.g. Blombos cave) to more inland sites (e.g. Sibudu, Rose Cottage, Melikane). Moreover, local variability in lithic assemblages during MIS 3 has been inferred to reflect fewer interactions between different groups/populations possibly driven by MIS 3 environmental uncertainty (Mackay et al., 2014). 410 However, the short time period covered by our record prevents any environmental-based conclusions and highlights the need for longer high-resolution paleoclimate reconstructions in the region.

### 5 Conclusions

This study presents a 3 kyr long, high-resolution and multi-proxy record of temperature and hydroclimate at the southern coast of South Africa during MIS 3 (45.2-48.3 ka). The Bloukrantz cave record suggests stable temperature throughout the 415 interval with an average temperature of 18.8 ± 0.5 °C, slightly warmer compared to the present day, whereas precipitation at the site appears to have been highly variable. Short episodes of higher $\delta^{18}O_c$, $\delta^{13}C_c$ and Sr/Ca values are likely linked to Prior Calcite Precipitation and to drier conditions. After 46 ka, a trend in the proxy baseline and a distinctly different isotope signal in the fluid inclusions is interpreted to reflect overall drier conditions with potential evaporation in the cave. Drier conditions between 46 to 43 ka are also observed in other records from the Summer Rainfall Zone and are matched by a 420 cooling at Dronning Maud Land in Antarctica, suggesting a potential influence of the Antarctic ice sheet, potentially through latitudinal displacement of the Southern westerly wind belt shifting the rain pattern over South Africa.

### Data availability

All results from this study are available in the appendix to this publication.



**Author Contribution**

Study design: JM, ANM, SEL; Methodology and data-acquisition: JM, TB, HAR, AFB, YK, JA, SA, ML; Visualization and original draft preparation: JM; Writing and editing: JM, ANM, SEL, TB, HAR, AFB, YK, JA, SA, ML.

**Competing interest**

The authors declare that they have no conflict of interest.

**Acknowledgements**

This work was funded by the Research Council of Norway through its Centres of Excellence funding scheme, SFF Centre for Early Sapiens Behaviour (SapienCE), project number 262618. Analyses were enabled by access to the national analytical infrastructure at UiB at Farlab (NFR grant number 245907) and EARTHLAB (NFR grant number 226171/F50); at the University of Bern the analytical work was funded by SNF grant numbers SNF-132646 (Stalclim), SNF-147674 (Stalclim II) and SNF-159563. We thank the South African Heritage Resources Agency (SAHRA), Heritage Western Cape (HWC) and

Cape Nature for granting the permits to enter the De Hoop Nature Reserve and collect speleothem samples for scientific analysis. We thank Ole Fredrik Unhammer, Magnus Mathisen Haaland, Sverre Asknes and Prof. Simon Armitage for their help with fieldwork. We thank Samantha Mienies at Wits University for curating the samples and helping with the permits. We thank Prof. Silvia Frisia at University of Newcastle for her help with the petrographic observations. We thank Prof. Harald Sodemann at UiB for his advice on the micro drop system. We thank Prof. Hai Cheng and Xuexue Jia for

supplementary U-Th dating performed during Covid-19 lockdown. We also thank Dr. Steffen Holzkämper, Prof. Karen Holmgren, Dr. Margit Simon and Dr. Brian Chase for respectively providing the Wolkberg cave speleothem data, the Lobatse cave speleothem data, the Tswaing crater data and the Cape Basin Record data.

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





**Figure 1: Map of southern Africa showing the study site (red circle) with the main rainfall zones (grey shading) and sites from the literature (yellow circles). Major atmospheric circulation is indicated by thin black arrows and major oceanic currents are indicated by thick white arrows.**

**BC - Blauwkrantz cave; WC - Wolkberg cave (Holkämper et al. 2009); LC - Lobatse cave (Holmgren et al. 1995); TC - Tswaing crater (Partridge et al. 1997). Marine cores MD02-2588 and CD154 17-17K (Simon et al. (2013). WRZ - Winter Rainfall Zone (dark grey); YRZ - Year-round Rainfall Zone (grey); SRZ - Summer Rainfall Zone (light grey); AC - Agulhas Current; BC - Benguela Current; SAC - South Atlantic Current.**






**Figure 2: Top) BL3 stalagmite. Dashed lines indicate the isotope transects. The black line indicates the hiatus (see text). Sr/Ca and isotopes analyses were measured on two different slabs resulting in slight offsets between the records (illustrated by the grey shadings in the bottom panel).**

**Bottom) Proxy data from BL3 vs depth. Top (black) - Sr/Ca as count-rate ratios from XRF scanning; middle (blue) - $\delta18Oc$; bottom (ochre) - $\delta 13Cc$. Grey shading indicates corresponding depths between the Sr/Ca and isotopic records. Dashed lines indicate the onset of darker layers in the stalagmite (c.f. top panel). Dating depths are indicated by the black square symbols on the x-axis (the open symbols correspond to the two samples measured at the Isotope Laboratory at Xi'an Jiaotong University).**




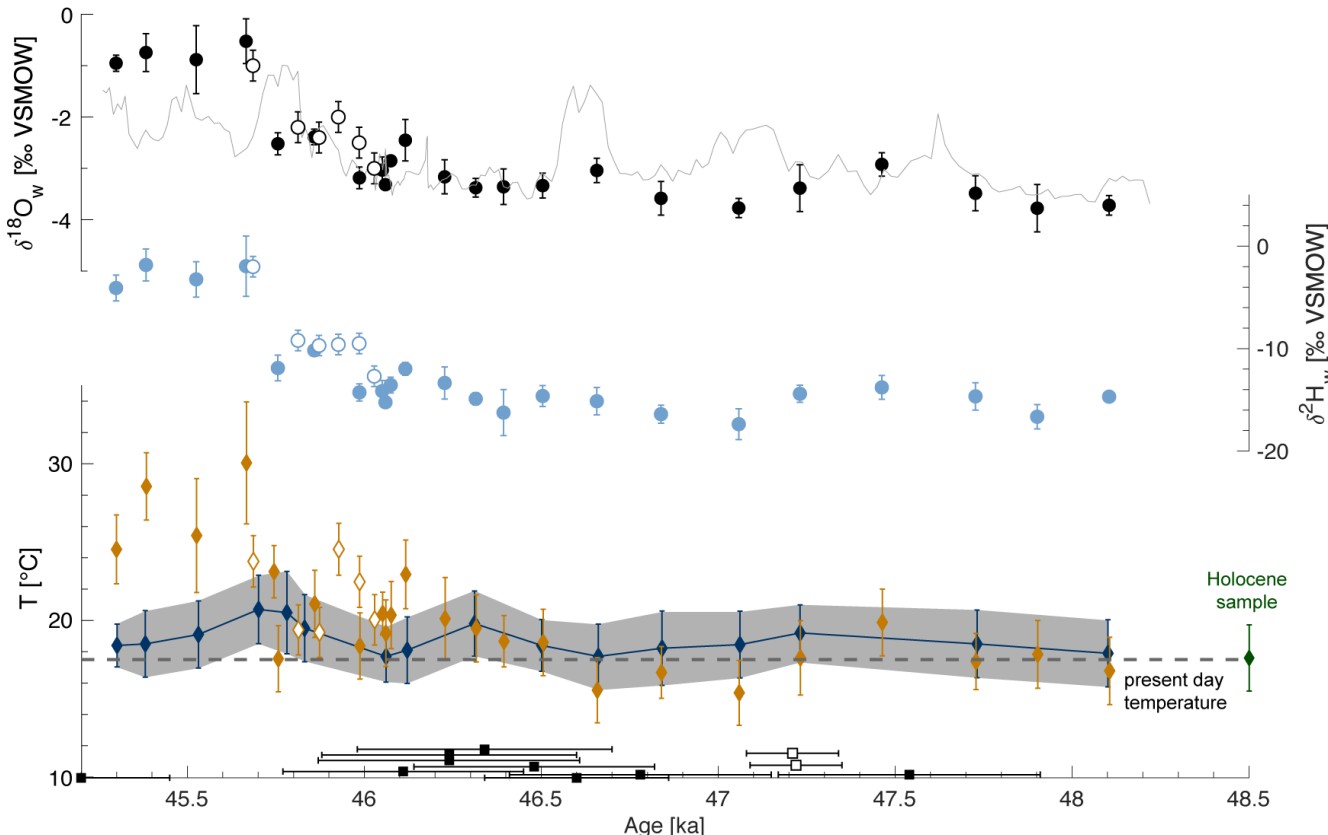

**Figure 3: BL3 proxy data vs age. Top: δ18Oc (grey line) and Foδ 18Ow (black circles); Middle: δ2Hw (light blue circles); Bottom:**
**Temperature reconstructions from fluid inclusion water isotope (ochre diamonds) and microthermometry (dark blue line and**
**diamonds.**

Open symbols correspond to samples analyzed at the University of Bern. In the bottom panel the dashed line indicates the present-
day temperature in the cave and the dark green diamond on the right corresponds to the topmost sample.

Black squares at the bottom indicates the U-Th dates and their associated error bars (2σ). The open square symbols correspond to
the two samples measured at the Isotope Laboratory at Xi'an Jiaotong University.





**Figure 4: Fluid inclusion water isotope data plotted in δ2Hw vs δ18Ow space (δ2Hw=4.79(±0.16)* 18Ow + 1.18(±0.47); R2=0.97). Lines indicate the global (black) and local (dashed blue – Cape Town) meteoric water lines. The color bar on the right indicates the water content for each sample. Open symbols correspond to samples analyzed at the University of Bern. Error bars ± 1σ.**



**Figure 5: Comparison to other climate records: a.** $\delta 18O$ from the EPICA Dronning Maud Land ice core in the Atlantic sector of the Antarctic Ice Sheet (EPICA community members 2006); **b.** $\delta 18O_c$ from Bloukrantz cave in the YRZ (this study); **c.** $\delta 13C_c$ from Wolkberg cave in the SRZ (Holzkämper et al. 2009); **d.** $\delta 13C_c$ from Lobatse cave in the SRZ (Holmgren et al. 1995); **e.** Annual precipitation reconstruction from Tswaing crater in the SRZ (Partridge et al. 1997); **f.** Ice Rafted Debris from core MD02-2588 in the southern Agulhas Plateau (Simon et al. 2013); **g.** Mean annual Sea Surface Temperature and Agulhas Leakage Fauna reconstructed from core CD154 17-17K in the south west Indian Ocean (Simon et al. 2013).