# Peer review of "Multi-proxy speleothem-based reconstruction of mid-MIS 3 climate in South Africa"

_Climate of the Past, 2023_

## Author Response (AR2)

General comments:

The manuscript "Multi-Proxy speleothem-based reconstruction of mid-MIS 3 climate in South Africa" presents new records of speleothem stable isotopes, strontium concentrations and temperature reconstructions based on fluid inclusion methods from Bloukrantz Cave, South Africa. In my opinion this manuscript addresses relevant scientific questions within the scope of Climate of the Past. Overall I think this manuscript would be a valuable publication in Climate of the Past after some major additions outlined below.

This is the first record of speleothem trace element compositions and fluid inclusion-based temperature reconstructions from this region. Since this is a very short and highly resolved record comparison to existing proxy records (which are usually much lower resolved) and the conclusions that can be reached are somewhat limited, nevertheless, I think this is a very relevant and important dataset and that the conclusions are substantial. The methods and assumptions used in the manuscript are valid and clearly outlined and their descriptions are sufficiently complete to allow for traceability of results. The results also support the interpretations and conclusions.

There is a clear distinction between previous work (including proper citations) and the addition of the current work. The title and abstract are clear and reflect the content of the article. I only have minor comments about the overall presentation and language (see below); symbols and abbreviations are correctly defined and used and references are appropriate except for a few minor additions/comments below.

Specific comments:

I have on major technical correction regarding the dating data tables in the supplementary materials. In the current form there is not enough information about the U-Th ages to judge their quality or to re-calculate them in the future e.g. if half-lives of some of the involved isotopes are updated. Ideally the following parameters (and their uncertainties) should be reported for each dating analysis: depth in sample (this is provided already), weight, uncorrected age result, 238U concentration, 232Th concentrations, 230Th concentration, 230Th/232Th ratio, 230Th/238U ratio, 234U/238U ratio, corrected age, reference (are ages before 1950, 2000, or relative to the year when the age was measured [give the year in this case]). - We agree that the reporting of the U/Th data can be improved. We will add the missing data in the revised documents following data reporting requirements (Dutton et al. 2017 doi:10.1016/j.quageo.2017.03.001).

A tab named U-Th data has now been added to the supplementary data.

I also have two main points that I think could be improved in the discussion

First, I think the section 'Hydroclimate reconstructions' is well written, but in it's comparison to other proxies it focusses a lot on very distant records in the summer rainfall region. The authors mention other speleothem records from the South African south coast in the Introduction and I understand that these records are very low resolved at the time when the sample used here was formed preventing a direct comparison. There are also two records from the Little Karoo that are not mentioned in the paper (Talma, A. S., & Vogel, J. C. (1992). Quaternary Research, 37(2),

203–213. https://doi.org/10.1016/0033-5894(92)90082-T; Chase, B. M. et al. (2021). Geology, 1. https://doi.org/10.1130/G49323.1). I think that something might might be learned from comparing the range of d18O and d13C values at the different caves sites – do they overlap or not and what might be the reasons? I also think that the Talma & Vogel (1992) record has a decent resolution for the Holocene section that overlaps with this new record and a comparison of that section might be feasible (the Chase record is a composite that mostly replicates the Talma & Vogel record in the Holocene). I think it is also worth mentioning that the pattern of the presented record with increasing stable isotope values during phases of global cooling is also found in previously published records (e.g. most of the published records from this region have somewhat higher values for d18O and d13C during cooler phases like MIS 4 than the warmer MIS 5), despite the very different resolution. The previously published records have been interpreted in a very different way, but this similarity in the relation to global change might suggest that the processes mentioned here also affected the other records.

We thank the reviewer for bringing these studies to our attention and for providing the Cango cave data. With regards to the MIS 3 section, the comparison is unfortunately limited by the resolution (see figure below, left column). The d18O values are higher compared to Bloukrantz cave, which can be explained by the higher altitude of Cango cave (~850 m) and the location further inland. The ranges in d13C values are more complicated to compare, as Bloukrantz cave speleothems are most likely influenced by PCP (based on the good correspondence between d13C and Sr/Ca we observe), which increases the d13C values compared to systems not influenced by PCP. This influence is, however, difficult to distinguish from other influences like vegetation change (C3 vs C4 plants) unless additional constraints are available.

[Figure]

The Holocene section of the Cango cave record presents a higher resolution and can hence be compared with the data from Bloukrantz cave (see figure, right column). In the Holocene part of the Bloukrantz cave record, we see strong covariation of both d18O and d13C with each other and also with Sr/Ca, similar to the MIS 3 section. This covariation suggests that PCP seems to exert the main control on the proxies also in the Holocene section of BL3, with values increasing with increasing PCP. We believe this to limit direct comparison of the absolute values with other regional records.

Regarding trends: In the Cango cave record, the d13C values are interpreted in terms of change in vegetation type (C3 vs C4 plants) as both vegetation density changes and PCP have been ruled

out (Chase et al. 2021). Both the Cango cave and Bloukrantz records display an increase in d13C values, interpreted as relative drier conditions at Bloukrantz cave and an increase in C4 plants at Cango cave associated with SRZ conditions dominance. Interestingly, the d13C increase correlates with an increase in reconstructed precipitation at Tswaing crater in the SRZ (Partridge et al. 1997). The observed drying at Bloukrantz cave could tentively be interpreted as decrease in Winter rain contribution and hence weaker westerlies (or southward displacement).
The Cango cave record displays significantly lower d18O values than the Bloukrantz cave record (as in the MIS3 part) with little variations.
Though these comparisons are indeed interesting, we believe including a discussion on the Holocene section would dilute the primary focus of our manuscript which is on the MIS 3 part, where we have significantly more data (temperature estimates from FIWI and Microthermometry). However, we agree that these records from Cango Cave, and their climatic control, need to be mentioned, and we will revise the text both in the introduction and the discussion to include them and other studies from Madagascar and Namibia.

We have added and mentioned some of the results from Cango cave in the revised manuscript (lines 67-71).

Second, I think the section on "Significance for the archaeological record" should be rephrased. I think this section is not very clear in its current form and it could be improved. E.g. The Howieson's Poort is a complex microlithic technology dating to late MIS 4 that can be found at many archaeological sites in southern Africa. It's disappearance at the beginning of MIS 3 and the decrease of site use intensity on the south coast may suggest a decrease of interactions between different hunter-gatherer groups and possibly a population decline and/or shift of activities towards inland locations. - We agree that this section's focus was not sufficiently clear. Our record covers a short (~3000 yrs) time period and is difficult to tie to the archeological records. We have thus decided to remove this short section from the current manuscript. We are in the process of working on a longer paleoclimate record where connections with the archeological record can be made more clearly.

The section 'Significance of the archeological record' has been removed.

Technical corrections:

Line 33-35: in sentence "Homo sapiens was anatomically modern as early as …" I think it is convention to italicize species names, also I would delete the words 'behavioral proxies in' later in this sentence - We will italicize *Homo Sapiens* and delete 'behavioral proxies'.

We have removed this sentence.

Line 38: "Episodes of significant cultural changes, …" there is an 'e.g.' before the second citation at the end of this sentence, either delete it or move it in front of the first one. – The 'e.g.' will be moved in front of the first citation.

The 'e.g.' has be moved in front of the first citation (lines 36).

Lines 68ff: "In the speleothem record, this is illustrated…" add some information and references to Talma & Vogel (1992) and Chase et al., (2021) here. - We will revise the text to add regional information and the mentioned references.

We have revised the text and included both Talma and Vogel (1992) and Chase et al. (2021) references (lines 67-71).

Line 85: "The theoretical background of this approach dates back to the 1960s…" – all the citations that follow are from the 40s' and 50s'- Indeed, we will revise the text.

We have corrected the years (line 86).

Line 114: Chapter 2.2 Sample description: the authors mention the calcite fabrics throughout this chapter, but only one image of the fabric is included in the Supplementary Materials. I think a few examples of the different fabrics mentioned here in the Supplement would be useful. - We will add figures of the different fabrics in the supplementary materials.

We have added Figure S4 in the supplement section illustrating the two calcite fabrics discussed in the manuscript.

Starting in Line 215: sections 3.2 Trace elements and 3.3 Stable isotopes as well as Figure 2: the text in the results describes the Sr/Ca record and both stable isotope records in terms of their temporal changes, yet, the figure that presents the results (Figure 2) plots them against depths. And the Sr/Ca and d13C records are not plotted against age in any of the figures. I think this should be harmonized, either by changing figure 2 to be plotted against age (the plot against depth could still be presented in the supplement if needed) or by referring to depths here (ages could be mentioned along with the depths maybe in parentheses). I also think that the order of Sr/Ca, d13C and d18O in Figure 2 from top to bottom should be the order in which they are mentioned in the text. - We agree that the figures and text should be consistent. We will keep figure 2 plotted against depth as it allows better to refer to the proxy transects (and related offsets). We will revise the text accordingly and add a figure in the supplement of the three proxies plotted against age. We will also make sure that the proxies are discussed in the order in which they are plotted.

We have kept figure 2 plotted against age and revised the text in sections 3.2 and 3.3 to add the depth when describing the results. We also now present the results in the order in which they appear in figure 2, Sr/Ca, d18O and d13C. We have added Figure S6 in the supplement where all three proxies are plotted against age.

Line 119: "After the hiatus, the Sr/Ca signal drops markedly and shows little variation with an average of 348,…" This sentence sounds like there is little change for the whole Holocene section and that the values stay below what they were before the hiatus. I don't think this is entirely true. I would rephrase to something like: "After the Hiatus, the Sr/Ca signal drops markedly and shows little variations with an average values of 348 between depths of 200 and 150 mm/ages of __ to__ ka. Between 150 and 70mm (__-__ka) values gradually increase to ~500. The top 70mm (__ to __ ka) show some of the highest variability of the record with averages around 500 (not sure about that value). - We agree, we will revise the manuscript and add a more thorough description of the proxies in the upper (post-hiatus) section.

We have revised sections 3.2 and 3.3 to add a more thorough description of the proxies in the upper part of the stalagmite.

Line 336 ff: "The inferred overall drying observed in our record…" I think the publication by Engelbrecht, F. A., et al. (2019. Quaternary Science Reviews, 226, 105879. https://doi.org/10.1016/J.QUASCIREV.2019.105879) could be cited here. They show that a northward shift of the westerlies during the LGM might reduce the amount of winter rainfall along a very narrow stretch of the south coast due to downwind effects along the Cape Fold Mountains. – We hank the reviewer for bringing this work to our attention. We will revise the manuscript including this reference and discussing some of its results.

We have added the results from Engelbrecht et al. (2019) in the text (lines 345 – 347).

Line 383: "Here, the water content displays little variation…" refer to Fig 4 along with Fig. S7 in this sentence. - We will revise the manuscript to refer to both Fig 4 and S7.

We now refer to both figures (lines 395).

Figures:

I have two general suggestions:

1. Remove the line breaks/paragraphs from the figure captions, I think it is not standard practice to do this. - We will. – We have removed the page breaks from the figure captions.
2. I would add subfigure denominations(a, b, c) to figures 2 and 3 instead of referring to 'top', 'middle' and 'bottom'. Especially in Figure 3 the authors refer to the bottom for the plot for ages as well as temperature reconstructions which is not very clear. I would advise the same for the supplementary figures. - We will add denominations for both figures 2 and 3.

We have deleted the page breaks from the figure captions and added letter denominations in figures 2, 3 and 5.

Line 217: Figure 1: blue shading indicates bathymetry of the surrounding oceans; is the current shown as SAC (South Atlantic Current) not the Antarctic Circumpolar Current? The SAC would be the section in the south Atlantic that also represents the southern branch of the subtropical gyre that then is deflected north into the BC, the ACC is what continues east. - Indeed, the current is wrongly labelled SAC. Figure 1 will be modified.

Figure 1 has been modified and SAC replaced by AAC.

Figure 2: Line 726: "Dashed Lines indicate the isotope transects." Add 'along the main growth axes of the speleothem' since in the text this is what they are said to indicate. - We will revise the text.

The text has been revised (line 737).

Line 730: "Dashed lines indicate the onset of darker layers…" It looks like the dashed lines were taken from the depth along the isotope profile but were not adjusted for the difference in depth in the Sr profile; are the depths of the dark layers known for the TE profile and could this be

adjusted similar to what is shown with the grey boxes? - Indeed, the dashed lines were not adjusted for the depth offsets between the TE and stable isotope profiles. We will correct this.

We have adjusted the dashed lines for the TE profile.

Line 744: Figure 4: in the text the authors often refer to the outlier data cluster in this plot and that they represent a specific age range. I think these outliers should be marked here, maybe just with a simple circle around the younger samples that are more offset from the meteoric water lines or by using different symbols for them? - We will indicate the cluster by a circle.

The outlier data cluster is now indicated by a red dashed circle (line 754).

Line 749: Figure 5: I think more recent versions of the EDML d18O record do not have the gap between 45 and 44 ka, see here: EPICA Community Members (2010): Stable oxygen isotopes of ice core EDML. doi:10.1594/PANGAEA.754444 - Indeed, thank you giving us the complete dataset reference. We will replot the EDML using the complete record.

Figure 5 has been replotted with the full EDML data set (a).

Supplements:

I recommend adding a title page to the file that states the title of the paper and names of authors and corresponding author. - We will add a title page.

We have added a title page to the supplement file.

Caption to Fig S2: correct 'Mars' to 'March'. - We will correct that.

We have added the title and corrected March in figure S2 caption.

**Citation**: https://doi.org/10.5194/cp-2023-1-RC1

**General comments**

In their manuscript Maccali and coauthors present a new multiproxy speleothem record of MIS3 climate variability in South Africa. The manuscript covers two bases, a climate record of AIM12, and a high-resolution comparison of two different fluid inclusion-based speleothem temperature proxies. The manuscript does very well not to 'fall between two stools' and covers both components well. The record itself is good, but relatively short, and while the growth phases show regional coherency and is worth remarking on, the climatic conclusions are necessarily limited by the time span. However, the additional comparison between microthermometry and fluid inclusion water isotope-based temperature reconstructions is strong and novel, and elevates the study. It is well worth consideration for Climate of the Past.

The novel result of the paper is the lack of significant temperature variability in the YRZ during AIM12, despite millennial (and centennial) scale hydroclimate variability. The result is disputed by the two temperature proxies, with FIWI method showing warming and the microthermometry showing no change within error. The authors reason that the microthermometry is more reliable, as the FIWI signal is influenced by enhanced evaporation due to the dry conditions. If true, then this marks a significant moment, where the relatively nascent microthermometry technique appears to outperform the more established FIWI method as it is less influenced by in-cave hydroclimate variability. I think the argument made in the manuscript for this being the case is reasonable as the temperature change using FIWI is unreasonably large.

My main issue with the result is the lack of presented consideration of nuance, error bars and reliability of microthermometry. The Indian Ocean cooling from 48-46ka is less than 2C while the mean-to-mean temperature warming of the microthermometry is as high as 3C. Therefore, absence of evidence for change (within error bars no change) here is not sufficient for the evidence of absence concluded by the paper. I'm not sure I would expect such a large temperature change in the subtropics to millennial scale variability, unless a major front was involved. This research group is doing tremendous work to improve microthermometry, but this result is perhaps overstated and needs nuance. – While we have aimed to make the uncertainty of the microthermometry method clear in text and figures, we agree that some of our statements can be read as implying complete absence of any temperature variations. We meant to say that we do not reconstruct a detectable change beyond the method's uncertainty, and notably not on the scale that would match the variability in d18O, d13C and Sr/Ca. We will modify our conclusions on the absence of temperature change accordingly.

We agree that the differences between data points are indeed not small, however these data are not as robust as in other cases (e.g. Løland et al. 2022) due to the very small amount of measurable inclusion. We have hence revised the text so it no longer states a complete absence of temperature variations, but rather the higher variability in the second half of the record (lines 242-247; 360ff)

The manuscript is excellently presented. It is well-written and concise, covering all major bases, with few errors. The number of technical corrections is very small. Congratulations.

After writing my review, I have also had the chance to look at the RC1 comments from Dr. Braun. They seem reasonable and I agree with the majority of them.

**Specific comments**

Should the title include reference to the microthermometry, as this is one of the highlights of the paper.- We prefer to keep our original title as we would like to highlight the complementarity of all the different methods rather than the merits of one method over the others.

Should prior calcite precipitation be changed to prior carbonate precipitation? Can prior calcite precipitation be demonstrated (calcite stalactite, U/Ca information)? - Indeed, we will change prior calcite precipitation for prior carbonate precipitation as we cannot demonstrate calcite precipitation.

We now refer to PCP as prior carbonate precipitation (line 279ff).

[Figure]

XRF scanning slab    Stable isotope slab

Was the XRF core scanning of the lower sections orientated perpendicular to the sampling axis of the stable isotopes in each individual growth phase, or just to the entire stalagmite. The former is not easy with an ITRAX and I would be curious as to how this was achieved. Was the stalagmite raised above the bed, was the bed adjusted, or was there enough room to reorientate the stalagmite? If the latter then by what technique was the data adjusted to the different depth scale, and could the authors comment to what extent was the data smoothed or otherwise compromised relative to the stable isotopes. - We think there was a misunderstanding; our XRF scans were performed parallel to the sampling axis, not perpendicular. The stalagmite was cut into flat slabs and further cut into smaller pieces. A flat support was inserted on the rail and the different pieces were placed on that support and rotated horizontally to align the isotope transects with the scanning direction, resulting in scans parallel to the isotope transects. The slabs were taped to prevent movement and adjusted for horizontality. The XRF scans and stable isotope transect were performed on different slabs, resulting in a slight depth offset, as illustrated in the inserted figure and figure 2 in the text. The data presented here are an average of three parallel scans performed a few mm apart. We will add these explanations to section 2.3.2 in the manuscript.

Further analytical explanations were added in section 2.3.2.

Paragraph starting Line 121: The MIS3 growth phase is mentioned here, for consistency the late Holocene growth phase should also be mentioned here, rather than wait until line 199. - We will mention the Holocene growth phase and the MIS 3 growth phase already here.

Both Holocene and MIS 3 growth phase are now mentioned (lines 122-123).

Line 185: I recommend moving the number of replicate measurements for microthermometry up from line 231 into the methods section. - We will move the replicate information into the method section.

Mention of the replicates is now in section 2.3.4 (lines 185-186).

Is the 'too old' outlier age excluded due to Uranium loss? Could that indicate any potential bias in microthermometry at those depths? - We lack evidence to unequivocally assess the cause of this 'too old' age. Both U-ratios and U-concentration for that sample are in the same range of values as the other 'closed-system' samples, making U-loss unlikely. In addition, it would require significant loss of U to alter the age while minor addition of $^{230}$Th could significantly impact the age (Borsato et al. 2003, Studi Trentini di Scienze Naturali, Acta Geologica, 80, 71-83). This sample might have a higher content of organic matter adsorbing $^{230}$Th and leading to that 'older' age. Such processes would however not impact the microthermometry data which we hence consider to be as reliable at this depth as throughout the rest of the record.

Line 283: Drip rate is not strictly the control on PCP, rather it is the measure. The control is the rate of infiltration through the karst and cave ceiling. – We will revise the manuscript accordingly.

We have revised the manuscript (line 291).

Line 291: Dissolution is a feature of undersaturation, which can be caused by very wet conditions. There is a need further supporting evidence of dry conditions. Dust is already included, but trends of proxies into may also help. – We agree with the reviewer. However the dissolution features were observed on top of organic layers (e.g. stromatolite-like structures) and could be explained by microbial activity during periods of lower drip rates allowing bacterial communities to colonize the speleothem surface. We will revise the text.

We have revised the text (lines 299).

Line 304: If unresolved hiatus are present then the duration of dry events could be even longer, not shorter. - We meant the duration of the isotopic peak could be shorter, but the duration of the dry event including both the isotopic peak and the growth stop would indeed be longer. We will revise the text to clarify this.

We have revised the text (line 313).

Figure 3/5: The Indian Ocean SST record should be shown alongside the microthermometry temperature reconstruction. It could either be in Figure 3 or Figure 5, depending on whether the authors view these as 'results and discussion' figures or 'temperature and hydroclimate' figures respectively. - We will consider adding the temperature reconstruction in Fig. 5 or add a figure in the supplementary material.

We have added the SST record along with d18Oc and temperature reconstruction in supplementary figure S10 and refer to it in the text (line 368).

Figure 3: With a good choice of colour and transparency, overlapping shaded error ranges for both FIWI and microthermometry should be possible. - We will try to show both FIWI and microthermometry error ranges in shaded coloring.

We have added the FIWI error as a shaded range.

Figure 5: There should be a better EDML age model through this interval. If not/alternatively, the Antarctic Temperature Stack (Parrenin, Science, 2013) and WAIS Divide (WDPM, Nature 2015) ice core records provide continuous Antarctic records through this period. - We will plot the complete EDML record (EPICA Community Members (2010): Stable oxygen isotopes of ice core EDML. doi:10.1594/PANGAEA.754444) as also suggested by reviewer 1.

We have now replotted figure 5 to include the complete EDML record.

Supplement: In the interests of transparency and open science, full age chemistry data should be reported. - These data were indeed missing; we will report the full dataset.

We have added a tab named 'U-Th data 'in the supplementary data excel file.

I think the authors are correct to go with an Antarctic dominated influence on regional climate variability. The 46.1 kyr BP and 45.5 kyr BP change seem to be suitably distant from Greenland millennial scale events (on the Buizert corrected INTIMATE chronology: GS13 (H5a) starts 48.59. GI12 starts 47.11, GS12 (H5) starts 44.51 kyr BP). The match of growth phase to AIM12 fits better, as does the onset of cooling with the Antarctic Temperature Stack (ATS).

Acknowledging the caveat that there is only so much one can determine from a single specimen, I wonder if there is room to comment on the growth periods of the BL3. 48.1 to 45.3 ka corresponds nicely to Antarctic Isotope Maximum 12. Further, this growth period fits very well with a speleothem growth interval in SW Madagascar at 47.9ka-43.6ka, attributed to the combined impact of high summer insolation and an Antarctic influence of Indian Ocean SSTs (Burns et al., 2022, QSR). There is a reasonable match also to a growth phase in Inland Namibia which starts at 47.3 ka, albeit one which lasts much longer (Railsback et al., Palaeo3, 2016, Railsback et al., QSR in review). Regional coherence of growth phases is suggestive of a genuine climate control, while the differences imply that the Antarctic millennial scale variability is more important at the more southerly latitudes of SW Madagascar and the YRZ, than in inland Namibia where the insolation control seems to be less sensitive to millennial variability. – We thank the reviewer for bringing these studies to our attention. The AIS influence on speleothem growth from Madagascar is interesting, and so is the growth in Namibia. We would like to mention here that on-going work on new speleothem samples collected recently indicates a continuous (except for potential shorter hiatuses) growth from ~90 to 45 ka which contrasts slightly with the discrete growth phases in Madagascar. We will revise the manuscript to include the studies of both Burns et al. 2022 and Railsback et al. 2016 and extend the regional comparison and the discussion on climatic control.

We have included these studies in the revised manuscript (lines 337-339; 325-326).

The Holocene growth phase really picks up around 3.7 ka, again approximately matching growth phases in inland Nambia (4ka onwards) and SW Madagascar (3.1 ka onwards)(Burns et al., QSR 2022, Faina et al. Malagasy Nature 2021), at the most recent summer insolation maxima. Maybe

this is too speculative, but it's a useful regional comparison of well-dated high-res records. - These studies indeed provide useful records for regional comparison during part of the Holocene. However, as we explain in our reply to reviewer 1, we prefer not to put more weight on the Holocene section to keep the main focus of the manuscript on the MIS 3 section.

**Technical corrections**

- Throughout: missing superscripting of 18, 13 and 2 for isotopes. - We will correct this. We have superscripted the number for isotopes.
- Line 8: Misspelling of Sciences - This will be corrected. – We have corrected the spelling.
- Line 54/62: SST should be defined at first occurrence. - This will be corrected. – SST is now defined line 57.
- Line 67: Specify timeframe of radiation. Mean annual solar/summer/etc. - We will add the missing information. – We have added the missing information (lines 62).
- Line 77: also include soil respiration processes as a major control of speleothem d13C - We will revise the text to include respiration processes. – The text has been revised (line 78).
- Line 164: "subscript c stands for calcite" has been mentioned before. - This was indeed mentioned line 80. We will remove it from line 164. – The text has been revised (line 80-81).
- Line 224: "As with the Sr/Ca record" - This will be corrected. – The text has been revised (line 227).
- Line 373: I recommend including the specific dates of these samples (46.7 and 47.7?) here. - We will add the specific dates of the samples. – We have revised the text to include the dates (line 385).
- Figure 5: The lines are very flat and do not 'show off' the data very well. Is there a way of making the y-axis variability more pronounced. Either by increasing the degree of overlap between panels and/or by making the figure narrower. - We agree and will try to increase the spread on the y-axis. – We have replotted figure 5 with an increased spread on the y-axis.

**Citation**: https://doi.org/10.5194/cp-2023-1-RC2